# MUDDFormer: Breaking Residual Bottlenecks in Transformers via Multiway Dynamic Dense Connections

**Da Xiao**[1]  **Qingye Meng**[2]  **Shengping Li**[2]  **Xingyuan Yuan**[2]

## Abstract

We propose MUltiway Dynamic Dense (MUDD) connections, a simple yet effective method to address the limitations of residual connections and enhance cross-layer information flow in Transformers. Unlike existing dense connection approaches with static and shared connection weights, MUDD generates connection weights dynamically depending on hidden states at each sequence position and for each decoupled input stream (the query, key, value or residual) of a Transformer block. MUDD connections can be seamlessly integrated into any Transformer architecture to create MUDDFormer. Extensive experiments show that MUDDFormer significantly outperforms Transformers across various model architectures and scales in language modeling, achieving the performance of Transformers trained with ~1.8×–2.4× compute. Notably, MUDDPythia-2.8B matches Pythia-6.9B in pretraining ppl and downstream tasks and even rivals Pythia-12B in five-shot settings, while adding only 0.23% parameters and 0.4% computation. Code in JAX and PyTorch and pre-trained models are available at https://github.com/Caiyun-AI/MUDDFormer.

## 1. Introduction

Residual connections (He et al., 2016), which help mitigate the vanish gradient problem, have become indispensable to training deep learning architectures from CNNs to Transformers, the latter becoming the de facto backbone for foundation models. Though being very simple and effective, residual connections still have limitations to be solved, especially with deep Transformers with dozens of layers made

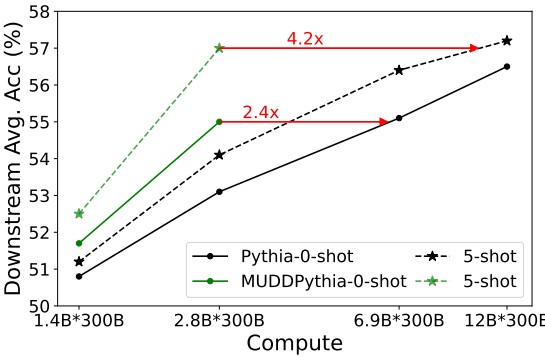

*Figure 1.* Downstream average accuracy of Pythia and MUD-DPythia with different sizes.

common by prevailing Transformer-based LLMs.

On one hand, although theoretical (Merrill et al., 2022) and experimental (Tay et al., 2021b) work have suggested that adding layers increases the expressive capacity and generalization performance of Transformers, it is observed that increasing depth beyond a certain point yields diminishing returns (Petty et al., 2023). The common practice of using Pre-Norm to stabilize training leads to the issue of representation collapse (Liu et al., 2020b), where hidden features in deeper layers become highly similar, and for popular families of open-weight LLMs, a large fraction of the layers can be removed with minimal degradation of performance (Gromov et al., 2024).

On the other hand, mechanistic interpretability studies reveal that Transformers do in-context learning tasks by composing model components (attention heads and MLPs) across different layers to form circuits, (Elhage et al., 2020; Wang et al., 2023; Merullo et al., 2024; Ni et al., 2025), where layers communicate with each other by writing to and reading from different subspaces of the residual stream. The residual stream as the shared communication channel may be overloaded and become the bottleneck for very deep models, hindering the formation of sophisticated circuits spanning distant layers necessary for complex tasks.

Dense connections (Huang et al., 2017) was proposed as a promising solution to the above issues, by allowing subsequent layers to directly access outputs of all preceding layers

[1]Beijing University of Posts and Telecommunications, Beijing, China [2]ColorfulClouds Technology Co., Ltd., Beijing, China. Correspondence to: Da Xiao <xiaoda99@bupt.edu.cn>.

(Figure 2 (a)). It has shown effectiveness for CNNs with DenseNet (Huang et al., 2017), for encoder-decoder Transformers with Deep Transformers (Wang et al., 2019) and for decoder-only Transformers with DenseFormer (Pagliardini et al., 2024). However, these dense connection approaches use static (either fixed (Huang et al., 2017) or learnable (Wang et al., 2019; Pagliardini et al., 2024)) dense connection weights that are shared across sequence positions and different input streams of a Transformer block. As will be shown, this rigidity severely limits their expressive capacity in Transformers.

In this work, we propose MUltiway Dynamic Dense (MUDD) connections, a simple yet effective approach to address the shortcomings of residual connections. Unlike existing dense connection approaches with static and shared connection weights, MUDD generates connection weights dynamically depending on hidden states at each sequence position and for each decoupled input (the query, key, value or residual) of a Transformer block. These weights are used by depth-wise aggregate modules to combine outputs from all preceding layers, creating multiple input streams for the current layer. MUDD connections can be seen as depth-wise multi-head attention (Vaswani et al., 2017) and the cross-layer communication bandwidth is expanded far beyond the restriction of the residual stream.

MUDD connections can be seamlessly integrated into any Transformer architecture to create MUDDFormer models. We conduct extensive experiments focusing on language model pretraining to evaluate MUDDFormer's effectiveness, efficiency and scalability. MUDDFormer significantly outperforms Transformer across various model architectures and scales (from 405M model on 7B tokens to 2.8B models on 300B tokens), achieving performance of Transformers trained with ~1.8×–2.4× compute. Notably, MUDDPythia-2.8B matches Pythia-6.9B in pretraining perplexity and downstream tasks and even rivals Pythia-12B in five-shot in-context learning settings (Figure 1), while adding only 0.23% parameters and 0.4% computation. We also evaluate MUDD connections on vision Transformers and analyze the trained models to elucidate why MUDD connections work.

## 2. Method

We begin with a standard Transformer decoder with $L$ layers on input sequence $X = \{x_0, ..., x_T\}$:

$$
\begin{aligned}
X_0 &= \text{Embedding}(X) \\
X_i &= \text{B}_i(X_{i-1}), \ i \in [1, L] \\
\text{Transformer}(X) &= X_L
\end{aligned} \quad (1)
$$

where $\text{B}_i(\cdot)$ is the $i$th Transformer block[1] composed of a multi-head attention (MHA) module followed by a fully

---

[1]In this paper we use "layer" and "block" interchangeably.

connected feed-forward network (FFN), both wrapped with Pre-LayerNorm (LN) residual connections:

$$
\begin{aligned}
X_\text{A} &= \text{MHA}(\text{LN}(X), \text{LN}(X), \text{LN}(X)) + X \\
\text{B}(X) &= \text{FFN}(\text{LN}(X_\text{A})) + X_\text{A}
\end{aligned} \quad (2)
$$

In this architecture, the output $X_i \in \mathbb{R}^{T \times D}$ ($D$ is model dim) of layer $i$ is used as input to layer $i + 1$. With *dense connections*, the input to layer $i + 1$ is an aggregation $\overline{X}_i \in \mathbb{R}^{T \times D}$ of outputs of *all* $i + 1$ preceding layers, from the embedding till layer $i$: $X_{:i} := \{X_0, ..., X_i\}$ (Figure 2 (a)). In this way, the cross-layer communication bandwidth is significantly increased compared to residual connections. To obtain Transformer with dense connections, we just add a *Depth-wise Aggregate* (DA) module after each layer to provide input for the next layer (cf. Eq. 1):

$$
\begin{aligned}
\overline{X}_0 &= X_0 = \text{Embedding}(X) \\
X_i &= \text{B}_i(\overline{X}_{i-1}); \ \overline{X}_i = \text{DA}_i(X_{:i}), \ i \in [1, L] \\
\text{DenseTransformer}(X) &= \overline{X}_L
\end{aligned} \quad (3)
$$

In the following subsections, we progressively derive Transformer with MUDD Connections (MUDDFormer), focusing on the computation inside the DA module after layer $i$.

### 2.1. Static Dense Connections

In the simplest case, the DA module aggregates previous layers' outputs by taking a weighted sum of them (Figure 2 (b), also equivalent to DenseFormer (Pagliardini et al., 2024)):

$$
\begin{aligned}
\overline{X}_i = \text{DA}_i^\text{static}(X_{:i}; \theta_i^s) &= \text{wsum}(\overset{i+1}{a_i}, \overset{(i+1) \times T \times D}{X_{:i}}) \\
&:= \sum_{j=0}^{i} \overset{1}{a_{ij}} \overset{T \times D}{X_j}
\end{aligned} \quad (4)
$$

where $\text{wsum}(\cdot)$ is the weighted sum function taking the sequences of weights and values as inputs. Scalar $a_{ij}$ is the $j$th value of the dense connection weight vector $a_i \in \mathbb{R}^{i+1}$ which is trainable parameters, i.e., $\theta_i^s = \{a_i\}$.

### 2.2. Dynamic Dense Connections

In Transformer-like sequence models, each layer processes information from multiple sequence positions and may benefit from differentiated and input-dependent dense connectivity for each position. Dynamic dense connections expand the connection weight for $X_j$ from a static scalar $a_{ij}$ to a vector $A_{ij} \in \mathbb{R}^T$, allowing $X_j$ to contribute differentially to each position $t \in [1, T]$ of $\overline{X}_i$ based on the hidden state $X_i[t] \in \mathbb{R}^D$ at that position. The $i + 1$ weight vectors stack into a matrix $A_i \in \mathbb{R}^{T \times (i+1)}$ which is generated dynamically by a function $\mathcal{A}_i(\cdot)$ depending on $X_i$ (Figure 2 (c), cf.

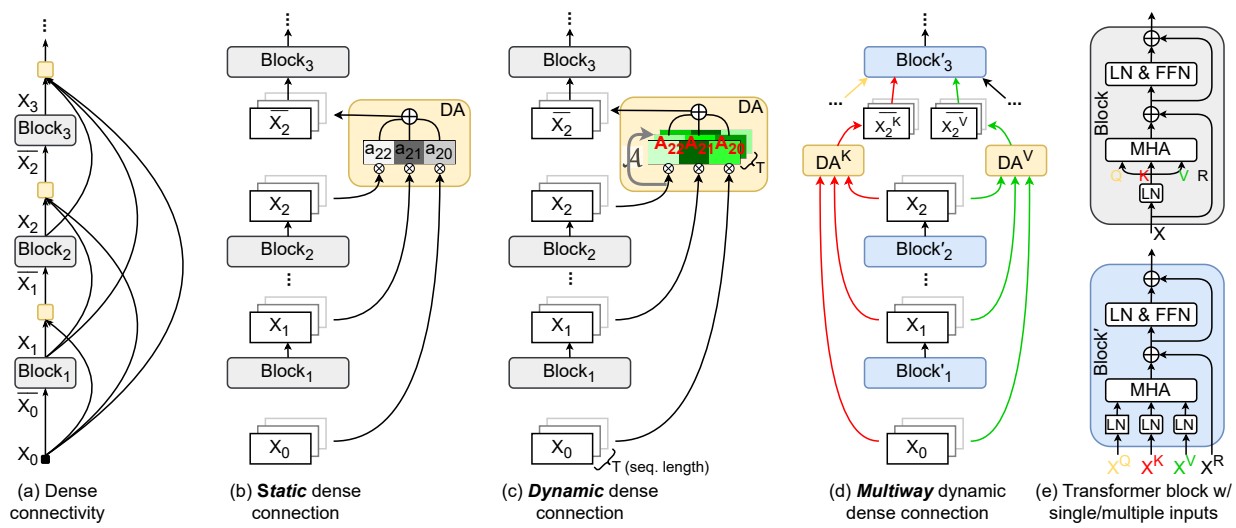

*Figure 2.* Architecture of Multiway Dynamic Dense Connections.

(a) Dense connectivity   (b) **Static** dense connection   (c) **Dynamic** dense connection   (d) **Multiway** dynamic dense connection   (e) Transformer block w/ single/multiple inputs

Eq. (4)):

$$\overline{X}_i = \mathrm{DA}_i^{\mathrm{dynamic}}(X_{:i}; \theta_i^d)$$
$$= \mathrm{wsum}(\overset{T\times(i+1)}{A_i} = \mathcal{A}_i(\overset{T\times D}{X_i}), \overset{(i+1)\times T\times D}{X_{:i}}) \quad (5)$$
$$:= \sum_{j=0}^{i} \overset{T\times 1}{A_{ij}} \odot \overset{T\times D}{X_j} \quad \text{(with broadcasting)}$$

where $A_{ij}$ is the $j$th column of $A_i$ (with a slight abuse of notation). We instantiate $\mathcal{A}_i : \mathbb{R}^D \to \mathbb{R}^{i+1}$ with an MLP parameterized by $W_1$ and $W_2$ which computes connection weights position-wise:

$$\mathcal{A}_i(X_i) = \mathrm{GELU}(\mathrm{RMSNorm}(X_i)W_1)W_2 + a_i \quad (6)$$

We apply RMSNorm to $X_i$ before MLP to stabilize training. We also add a static weight vector $a_i$ acting as learnable prior for dense connectivity. The trainable parameters are $\theta_i^d = \{W_1 \in \mathbb{R}^{D\times(i+1)}, W_2 \in \mathbb{R}^{(i+1)\times(i+1)}, a_i \in \mathbb{R}^{i+1}\}$.

Overall, dynamic dense connections can be viewed as a form of depth-wise single-headed self-attention (Vaswani et al., 2017) with query $X_i$ and keys $X_{:i}$, and the attention weights are computed from the query side[2].

### 2.3. Multiway Dynamic Dense Connections

In a Transformer block, a single input is reused simultaneously as the query, key, value and residual of the MHA module (Figure 2 (e) top). These input streams play divergent roles and we hypothesize that they benefit from differentiated dense connectivity. To enable this, we first turn a normal Transformer block $\mathrm{B}(X)$ into a multi-input one $\mathrm{B}'(X^Q, X^K, X^V, X^R)$ by *decoupling* its input into four

streams for query, key, value and residual, respectively (Eq. (7), Figure 2 (e) bottom), and then instantiate four DA modules, each specializing in one stream's dense connectivity (Eq. (8), Figure 2 (d))[3]:

$$X_A' = \mathrm{MHA}(\mathrm{LN}(X^Q), \mathrm{LN}(X^K), \mathrm{LN}(X^V)) + X^R$$
$$\mathrm{B}'(X^Q, X^K, X^V, X^R) = \mathrm{FFN}(\mathrm{LN}(X_A')) + X_A' \quad (7)$$

$$\overline{X}_0^Q = \overline{X}_0^K = \overline{X}_0^V = \overline{X}_0^R = X_0 = \mathrm{Embedding}(X)$$
$$X_i = \mathrm{B}_i'(\overline{X}_{i-1}^Q, \overline{X}_{i-1}^K, \overline{X}_{i-1}^V, \overline{X}_{i-1}^R);$$
$$\overline{X}_i^Q, .., \overline{X}_i^R = \mathrm{DA}_i^Q(X_{:i}), .., \mathrm{DA}_i^R(X_{:i}), \ i \in [1, L] \quad (8)$$
$$\mathrm{MUDDFormer}(X) = \overline{X}_L^R$$

By making the dense connections multiway, the cross-layer communication bandwidth is further increased significantly. Multiway dynamic dense connections can be seen as depth-wise multi(4)-head attention. This vertical cross-layer attention can be composed with the horizontal cross-token attention in Transformer to form pathways adaptively, enhancing information flow across the whole model when performing in-context learning tasks.[4] At this point, we finally obtain MUDDFormer by integrating static, dynamic and multiway dense connections. Complete pseudo-code for MUDDFormer is given in Appendix B.

### 2.4. Parameter Re-allocation

Due to the dense connections, MUDDFormer's upper layers have the opportunity to process more information than lower layers and thus may need more parameters. We re-allocate the parameters of a standard Transformers to make the size

---

[2]For further explanation, see Appendix A

[3]This is a logical view for clarity. In practice, these DSs can be combined for efficiency. See pseudocode in Appendix B.

[4]For an illustrative example of this, see Figure 7.

of FFN sub-layers grows with depth. Specifically, let $D_f$ be the hidden dim of the original FFN, we compute $D'_f(i)$, the hidden dim of FFN at layer $i$ for MUDDFormer using linear interpolation:

$$D'_f(i) = \frac{0.5(L - i) + 1.5(i - 1)}{L - 1} D_f \qquad (9)$$

i.e., the FFN hidden dim $D'_f$ grows linearly from $0.5D_f$ to $1.5D_f$. The total number of parameters remains unchanged.

## 2.5. Optional Normalization

To stabilize training models with large depth/width ratios, we propose a variant of MUDDFormer by applying RM-SNorm before and after DA module, and adding a residual connection to DA module *after* the post-RMSNorm:

$$\begin{aligned}
X_{:i} &= \{\mathrm{Norm}(X_0), ..., \mathrm{Norm}(X_i)\} \quad \text{(PreDANorm)} \\
\overline{X}_i &= \mathrm{Norm}(\mathrm{DA}_i(X_{:i})) + X_i \qquad \text{(PostDANorm)}
\end{aligned} \qquad (10)$$

It is similar to the hybrid-norm strategy used by recent models such as Gemma 2 (Team et al., 2024b) and Grok-1 (xai org, 2024), though we apply it to DA modules instead of MHA/MLP modules. We use this PrePostDANorm variant to train the DeepNarrow models in scaling law experiments in Section 3.1 and the MUDDViT model in Appendix F.

## 2.6. Complexity Analysis

Table 1 shows the ratios of extra parameters and computation introduced by MUDD connections with both analytical results and typical concrete values. The derivations are in Appendix C. The ratio of extra parameters, i.e. parameter count of $W_1$ and $W_2$ of DA modules divided by that of the whole model, is proportional to the rectified depth/width ratio $\eta = \frac{L+3}{D}$. The ratio of extra computation, i.e. FLOPs of generating MUDD connection weights and cross-layer aggregation divided by FLOPs of the whole forward pass, besides proportional to $\eta$, decreases with $\rho = \frac{T}{D}$. Both ratios are negligible for commonly used settings.

*Table 1.* Ratios of extra parameters and computation introduced by MUDD connections: (last row) analytical results and (upper rows) concrete values for typical model architectures and hyperparameters. L = number of layers, T = sequence length.

| Model Size | $R_{\Delta\text{params}}$ | $R_{\Delta\text{FLOPs}}$ | L | D | T | $\eta = \frac{L+3}{D}$ | $\rho = \frac{T}{D}$ |
|---|---|---|---|---|---|---|---|
| 1.4B | 0.22% | 0.38% | 24 | 2048 | 4096 | 0.0132 | 2 |
| 1.34B | 0.49% | 0.8% | 42 | 1536 | 4096 | 0.0293 | 2.67 |
| 2.8B | 0.23% | 0.4% | 32 | 2560 | 4096 | 0.0137 | 1.6 |
| 6.9B | 0.14% | 0.26% | 32 | 4096 | 4096 | 0.0085 | 1 |
| Formula | $\frac{\eta}{6}$ | $\frac{\eta}{3 + \rho/4}$ | | | | | |

# 3. Experiments

**Implementation Details** We implement MUDDFormer model and training in JAX. We initialize the MUDD connection weight generating parameters $W_1$ and $W_2$ with $\mathcal{N}(0, \frac{1}{D})$ and 0 respectively, and initialize the static weight vector $a_i$ with 1 at $a_{ii}$ and 0 elsewhere. This reduces MUDDFormer to Transformer at the beginning of training, which is found to be critical for good performance. If PrePostDANorm is used, we initialize the scale parameters of Pre-DA and Post-DA RMSNorms with 1 and 1e-3, respectively, and initialize $a_i$ to 0 because $X_i$ is added as the residual after DA modules (Eq. 10). For the other parameters outside DA modules, we use Xavier normal initializer.

**Organization** Our evaluation focuses on language modeling with decoder-only Transformer architecture, analyzing both pretraining loss scaling laws (Section 3.1 and Section 3.2) and downstream task performance (Section 3.3) with large scale training on the Pile dataset (Gao et al., 2020). Section 3.4 elucidates why MUDDFormer works through analyzing trained models, followed by efficiency analysis (Section 3.5) and ablations (Section 3.6). Extended vision experiments are provided in Appendix F.

## 3.1. Scaling Laws

**Settings** Table 2 (top half) specifies the model sizes and hyperparameters for scaling experiments, which are mostly taken from GPT-3 specifications (Brown et al., 2020). We untie input and output embedding matrices. We train with context length 2048 and set the number of training tokens to roughly match Chinchilla scaling laws (Hoffmann et al., 2022). The other hyperparameters are in Appendix D.1. In another set of scaling experiments we trade off some width for depth to train *DeepNarrow* models (Table 2 (bottom half)) to see if MUDD connections offer any benefits for this style of scaling.

*Table 2.* Model sizes and hyperparameters for scaling experiments.

| params | $n_{\text{layers}}$ | $d_{\text{model}}$ | $n_{\text{heads}}$ | learning rate | batch size (in tokens) | tokens |
|---|---|---|---|---|---|---|
| 405M | 24 | 1024 | 16 | 3e-4 | 0.5M | 7B |
| 834M | 24 | 1536 | 24 | 2.5e-4 | 0.5M | 15B |
| 1.4B | 24 | 2048 | 32 | 2e-4 | 0.5M | 26B |
| *Scaling by depth* | | | | | | |
| 797M | 34 | 1280 | 20 | 2.5e-4 | 0.5M | 15B |
| 1.34B | 42 | 1536 | 24 | 2e-4 | 0.5M | 26B |

**Baselines** We compare MUDD with two recently proposed approaches to enhancing residual connections in Transformers: DenseFormer (Pagliardini et al., 2024) (same as the static dense connections described in Section 2.2) and Hyper-Connections (Zhu et al., 2025). We also compare to

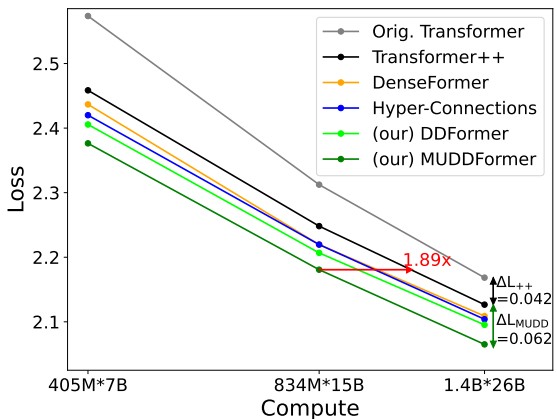

*Figure 3.* Scaling curves of MUDDFormer and baseline models.

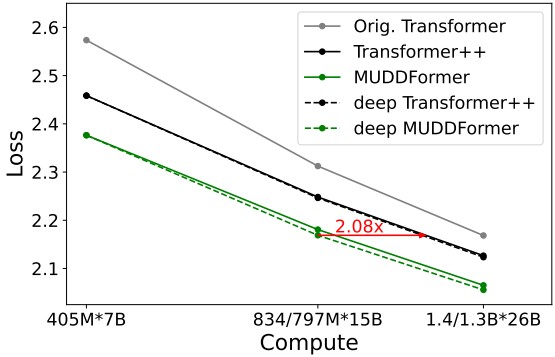

*Figure 4.* Depth scaling of MUDDFormer and Transformer++.

Transformer with dynamic dense connections (DDFormer) as described in Section 2.2. All these approaches are applied to an improved and now widely adopted Transformer architecture (Touvron et al., 2023) with rotary positional encoding (RoPE) (Su et al., 2024), SwiGLU MLP (Shazeer, 2020), etc. (often called Transformer++). We also include the plot for the original Transformer architecture used by GPT-3 as a comparison. The details for these baselines are in Appendix D.2.

**Results** Figure 3 plots Pile validation loss scaling curves of the models. While DenseFormer and Hyper-Connections show clear advantage over Transformer++, DDFormer outperforms them by adding dynamicity to dense connections. MUDDFormer further improves upon DDFormer by making the dense connections multiway, significantly outperforming all baselines on models ranging from 405M to 1.4B. It can be estimated that MUDDFormer-834M matches the loss of Transformer++ trained with $1.89\times$ compute.

Figure 3 also shows that as an architectural improvement, MUDDFormer's gain over Transformer++ ($\Delta L_{\text{MUDD}}$) remains stable while scaling, exceeding Transformer++'s own gain over Transformer ($\Delta L_{++}$) beyond 834M parameters. This shows the favorable scalability of MUDDFormer, particularly considering that Transformer++ has incorporated

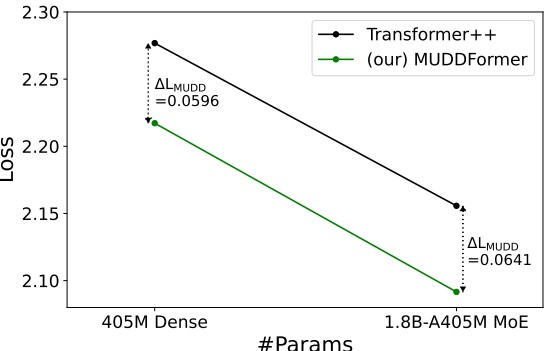

*Figure 5.* MUDDFormer with dense vs MoE models.

major architectural improvements (RoPE, SwiGLU MLP, etc.) over original Transformer since its invention.

Figure 4 demonstrates MUDDFormer's enhanced *depth utilization*: while Transformer++ shows diminishing returns beyond 24 layers (almost coincident scaling curves), MUDDFormer DeepNarrow maintains gains up to 42 layers. This validates that MUDD connections alleviate depth-induced bottlenecks by enhancing cross-layer information flow.

### 3.2. MoE Models

**Settings** Transformer with Sparse Mixture-of-Experts (MoE) and MUDDFormer are both architectures with dynamic weights. MoE uses these weights to select experts *within* a layer while MUDDFormer uses the weights to aggregate outputs *across* layers. They are complementary approaches and can be combined. To empirically compare MoE and MUDDFormer, we train a Transformr++ MoE model 1.8B-A405M with the same activated parameters as the 405M dense model, using the same training settings in the scaling law experiments. For MoE specific settings, we largely follow OLMoE (Muennighoff et al., 2025), using dropless token-choice routing to choose 2 experts out of 16 for each token with expert hidden dim 1408. The FLOPs of the MoE model is nearly identical to that of the dense model. We apply MUDD connections to both the dense and MoE models. All the models are trained on 26B tokens.

**Results** As shown in Figure 5, compared to the same Transformer++-405M, MUDDFormer-405M achieves ~50% loss reduction of Transformer++-1.8B-A405M MoE, which is $4.4\times$ larger. MoE works by *decoupling* parameters and computation and relies on significantly expanding parameters for good performance, in contrast to MUDDFormer's efficient utilization of *both* parameters and computation. More notably, MUDD demonstrates *more* benefits for the MoE model ($\Delta$loss $-0.0641$) than for the dense model ($\Delta$loss $-0.0596$), a result of particular practical significance given the increasing adoption of MoE architectures in frontier LLMs (Team et al., 2024a; Liu et al., 2024b; Team, 2025; Yang et al., 2025).

*Table 3.* Zero-shot and five-shot downstream evaluations results.

| Model | Pile ppl↓ | FLAN ppl↓ | LAM BADA | PIQA | Wino Grande | ARC -E | ARC -C | SciQ | Logi QA | BoolQ | Hella Swag | RACE -M | RACE -H | Avg acc↑ /Δacc |
|---|---|---|---|---|---|---|---|---|---|---|---|---|---|---|
| | | | *0-shot* | | | | | | | | | | | |
| Pythia-1.4B | 7.29 | 9.30 | 61.6 | 71.0 | 57.2 | 60.5 | 26.1 | 86.6 | 21.4 | **63.3** | 40.5 | 37.3 | 33.9 | 50.8 |
| **MUDDPythia-1.4B** | **6.92** | **8.54** | **63.9** | **71.8** | **57.4** | **61.6** | **26.2** | **87.2** | **23.0** | 62.0 | **42.6** | **38.7** | **34.7** | **51.7/+0.9** |
| Pythia-2.8B | 6.63 | 8.16 | 64.7 | 73.9 | 59.4 | 64.4 | 29.5 | 88.2 | 21.2 | 64.5 | 45.4 | 38.1 | 34.9 | 53.1 |
| **MUDDPythia-2.8B** | **6.29** | **7.50** | **68.5** | **74.6** | **61.4** | **66.5** | **31.9** | **90.4** | **21.5** | **68.1** | **46.8** | **39.0** | **36.7** | **55.0/+1.9** |
| Pythia-6.9B | 6.29 | 7.85 | 67.3 | 75.2 | 60.9 | 67.3 | 31.3 | 89.7 | 25.3 | 63.7 | 48.0 | 40.6 | 37.0 | 55.1 |
| Pythia-12B | 6.01 | 7.26 | 70.5 | 76.0 | 63.9 | 70.2 | 31.8 | 90.2 | 22.4 | 67.4 | 50.3 | 40.6 | 38.3 | 56.5 |
| **MUDDFM-2.8B** | 6.01 | 7.08 | 70.7 | 75.7 | 63.4 | 70.4 | 34.2 | 91.8 | 24.0 | 67.4 | 49.5 | 40.6 | 38.1 | **56.9** |
| | | | *5-shot* | | | | | | | | | | | |
| Pythia-1.4B | - | - | 54.5 | 71.0 | 57.5 | 63.1 | **28.9** | 92.2 | 22.9 | **63.0** | 40.5 | 35.4 | 34.6 | 51.2 |
| **MUDDPythia-1.4B** | - | - | **58.2** | **73.0** | **59.0** | **64.1** | 28.2 | **94.0** | **23.8** | 61.5 | **42.6** | **37.9** | **35.2** | **52.5/+1.3** |
| Pythia-2.8B | - | - | 60.5 | 73.6 | 60.6 | 67.3 | 32.3 | 94.3 | 21.7 | 65.6 | 45.1 | 38.4 | 35.6 | 54.1 |
| **MUDDPythia-2.8B** | - | - | **63.6** | **75.5** | **63.6** | **70.3** | **34.0** | **95.5** | **28.1** | **67.5** | **47.1** | **44.5** | **37.3** | **57.0/+2.9** |
| Pythia-6.9B | - | - | 63.8 | 75.5 | 63.7 | 70.2 | 35.6 | 95.1 | 27.0 | 65.7 | 48.1 | 39.0 | 36.5 | 56.4 |
| Pythia-12B | - | - | 67.3 | 76.0 | 64.2 | 71.0 | 36.5 | 95.3 | 21.8 | 68.0 | 50.3 | 40.1 | 38.8 | 57.2 |
| **MUDDFM-2.8B** | - | - | 65.6 | 76.4 | 66.8 | 73.0 | 39.2 | 95.6 | 25.2 | 70.9 | 49.8 | 41.4 | 38.0 | **58.4** |

## 3.3. Large Scaling Training and Downstream Evaluations

**Settings** We compare MUDDFormer with the open source Pythia model suit (Biderman et al., 2023) at large scale training on 300B tokens of Pile. Specifically, we train two models, MUDDPythia-1.4B and MUDDPythia-2.8B, and compare them with Pythia models ranging from 1.4B to 12B. For fair comparison and clear quantification of the gain brought by MUDD, except adding MUDD connections as described in Section 2, MUDDPythia uses exactly the same architecture choices (e.g. parallel attention and MLP, rotary embedding with 1/4 head dim) and training hyperparameters (e.g. optimizer settings, learning rate schedule, batch size, context length, initialization methods) as Pythia (refer Biderman et al. (2023) Appendix E for details). To evaluate if MUDD connections also work well with more advanced Transformer++ architecture and training recipe at this large scale, we also train MUDDFormer-2.8B based on Transformer++ instead of Pythia architecture with a larger learning rate of 3.2e-4 cosine decayed to 3.2e-6. Except these two changes, the other architectural and training hyperparameters are kept the same as MUDDPythia-2.8B.

**Evaluation Datasets** Besides the datasets used by Pythia for downstream evaluation (LAMBADA (Paperno et al., 2016), PIQA (Bisk et al., 2020), WinoGrande (Sakaguchi et al., 2021), ARC (Clark et al., 2018), SciQ (Welbl et al., 2017), LogiQA (Liu et al., 2020a)), we also include BoolQ (Clark et al., 2019) and HellaSwag (Zellers et al., 2019) for commonsense reasoning, RACE (Lai et al., 2017) for reading comprehension, all of which are widely used benchmarks. We evaluate zero-shot and five-shot results using

LM evaluation harness (Gao et al., 2023).

**Results** As shown in Table 3 and Figure 1, besides lower Pile validation ppl, MUDDPythia also significantly outperforms Pythia at 1.4B and 2.8B scales on downstream task accuracies. Notably, MUDDPythia-2.8B matches Pythia-6.9B (2.46× compute) on both pretraining ppl and downstream evaluation. Augmented with better Transformer++ architecture and training recipe, MUDDFormer-2.8B even outperforms Pythia-12B.

Table 3 also reports target span perplexities on a randomly sampled subset of the FLAN Collection dataset (Longpre et al., 2023), which features data of instructing following, chain-of-thought, in-context few-shot learning, etc. The advantage of MUDDPythia on FLAN is even larger than on Pile with MUDDPythia-2.8B significantly outperforming Pythia-6.9B, showing that MUDD connections have more advantage in improving these valued emergent abilities of LLMs (Wei et al., 2022). The enhanced in-context learning ability is also manifested by the larger Δaccuracies of 5-shot results compared to 0-shot (e.g. 2.9% vs. 1.9%), where MUDDPythia-2.8B is on par with Pythia-12B (4.29× compute). The multiway dense connections applied to the query, key, and value streams of MHA modules are likely to enhance the functionality of attention heads, which are crucial for in-context learning (also see analysis in Section 3.4).

Finally, the gains of MUDD connections with the 2.8B model are larger than those with the 1.4B model in both zero-shot and five-shot evaluations, further demonstrating the scalability of MUDD connections.

### 3.4. Analyzing and Understanding

We analyze and compare Pythia-2.8B and MUDDPythia-2.8B trained in Section 3.3 to elucidate why MUDD connections work. The analysis is done on 1024 randomly sampled sequences of length 2048 from Pile validation set.

**Representation Collapse** Figure 6 quantifies representation collapse through cosine similarity between the inputs of adjacent layers. While Pythia exhibits progressive collapse with >0.97 similarity in later layers, MDDDPythia maintains more distinct input representations, particularly in the value stream. Thanks to input stream decoupling and stream-specific aggregation, DA modules can freely aggregate distinct value input streams for MHA modules of each layer at each sequence position. These values will then be moved to *other* positions by MHA at this layer without polluting the residual stream of the *current* position. The relative importance of dense connections for the value stream is also evidenced by ablation studies (Section 3.6). We compare illustrative V-composition circuits[5] in Transformer and MUDDFormer in Figure 7 to highlight the benefit of MUDD connections and input stream decoupling, which results in more direct and cleaner information pathways. MUDD has similar effect on Q/K-composition circuits.

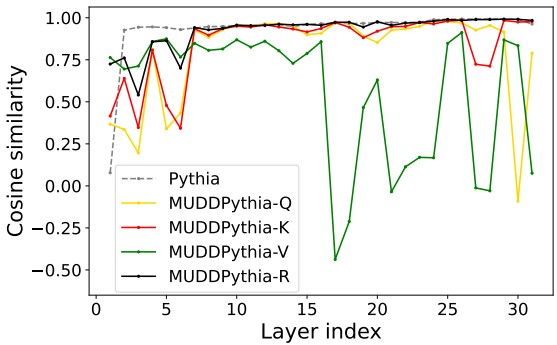

*Figure 6.* Cosine similarity between the inputs of the current layer and the preceding layer.

**Attention Head Activation** Transformer models often exhibit *null attention* (Vig & Belinkov, 2019) where attention heads focus on the initial tokens or some special tokens as default "attention sinks" (Xiao et al., 2024b) when no relevant tokens are found. We define a head to be *active* at a position if its maximum attention weight *does not* fall on the first two positions of the sequence or on tokens "<bos>", "." and "\n". We compute the activation ratio of attention heads for each layer by averaging over all heads of that layer and all sequence positions and plot the results in Figure 8. In Pythia, most heads remain inactive beyond the first few layers, limiting their contribution. MUDDPythia demonstrates ~2.4× higher activation ratio across layers, particularly in

---

[5]common and important in many tasks, e.g. Wang et al. (2023); Ni et al. (2025). For introduction, refer to Elhage et al. (2020).

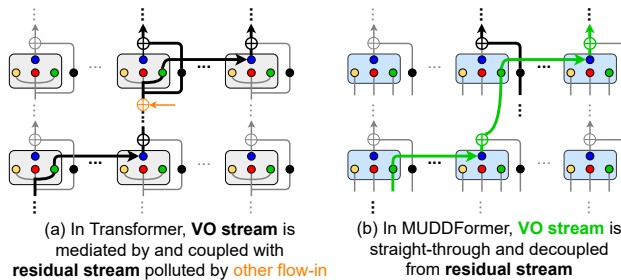

(a) In Transformer, **VO stream** is mediated by and coupled with **residual stream** polluted by other flow-in

(b) In MUDDFormer, **VO stream** is straight-through and decoupled from **residual stream**

*Figure 7.* Illustrative V-composition circuits in Transformer vs. in MUDDFormer. Colored circles are MHA's inputs of the query (yellow), key (red), value (green) and residual (black) streams and output (blue). LayerNorms and MLPs are omitted.

deeper layers.[6] This vitalization of attention heads stems from the multiway dense connections on the Q/K/V streams of MHA modules, ultimately improving in-context learning.

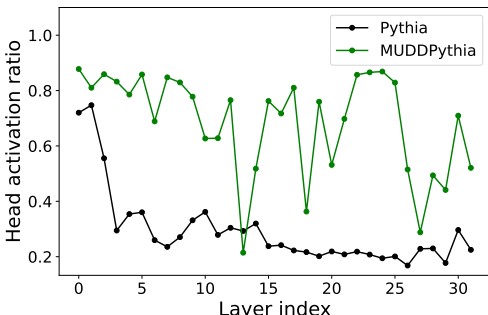

*Figure 8.* Attention head activation ratio by layers.

### 3.5. Training and Inference Efficiency

Besides theoretical complexity analysis in Section 2.6, we assess training and inference efficiency of MUDDFormer compared with Transformer in real-world settings.[7]

**Settings** Though we use Pythia's architecture in large scale training, the evaluation in this section is done on untrained models with Transformer++ (Llama) architecture, which is of more practical relevance due to better performance. We measure on three model sizes: 1.3B, 2.8B and 6.9B, which are "Llama-ed" version of Pythia 1.4B, 2.8B and 6.9B respectively. We train on Google Cloud TPU v5p-128 pods with context length of 2048, batch size of 2M tokens and measure training throughput. We do inference on NVIDIA A100 80G GPU with prompt length of 4096, batch size of 1 and measure the speed to generate 128 tokens. We repeat the measurements 3 times and take the average. We implement training and inference in pure JAX and PyTorch respectively

---

[6]Visualization of sampled attention patterns for heads in Pythia and MUDDPythia are shown in Appendix G.

[7]For additional results of analytical and measured memory usage and training wall-clock time of MUDDFormer vs Transformer, see Appendix E.

*Table 4.* Training throughput and inference speed comparison between Transformer++ and MUDDFormer.

| Model Size | Training (K tokens/s) | | Inference (tokens/s) | |
|---|---|---|---|---|
| | TFM++ | MUDDFM | TFM++ | MUDDFM |
| 1.3B | 1147 | 1030 ×89.8% | 325 | 286 ×88.1% |
| 2.8B | 684 | 575 ×84.0% | 181 | 163 ×90.0% |
| 6.9B | 332 | 318 ×95.6% | 95.5 | 89.7 ×94.0% |

without writing any Pallas/Triton-based custom kernels. We use torch.compile (PyTorch 2.5.1) to accelerate both Transformer++ and MUDDFormer.

**Results** As shown in Table 4, the training and inference overheads, while larger than the theoretical estimates in Table 1 and not negligible, are entirely acceptable considering the significant performance gain. The overheads primarily stem from the series of small operations and additional I/O introduced by DA modules. We believe that kernel fusion techniques offer potential for further acceleration and leave it for future work.

### 3.6. Ablations and Variants

We conduct ablation studies with the 405M Transformer++/MUDDFormer models in scaling law experiments for language modeling in Section 3.1.

**Ablation settings** We do two groups of experiments and report the perplexity results in Table 5. In the first group, we progressively add the four components, i.e. static (Section 2.1), dynamic (Section 2.2), multiway (Section 2.3) dense connections and parameter re-allocation (2.4)) to Transformer++ to finally obtain MUDDFormer to compare the contribution of each component. In the second group, we focus on the multiway aspect and study the effect of dense connections for the four decoupled inputs streams by replacing each of them with normal residual connection respectively.

**Results** All three ingredients of dense connections, i.e. *static*, *dynamic* and *multiway*, make contributions. While parameter re-allocation is effective on MUDDFormer, it *deteriorates* Transformer++. Removing dense connections for each of the four streams hurts performance and the value stream benefits most from dense connections.

*Table 5.* Ablations of MUDDFormer's components.

| Config | ppl | Config | ppl |
|---|---|---|---|
| Transformer++ | 11.68 | MUDDFormer | **10.77** |
| +Static Dense | 11.44 | −Q dense | 10.89 |
| +Dynamic Dense | 11.09 | −K dense | 10.90 |
| +Multiway Static Dense | 11.27 | −V dense | 11.05 |
| +Multiway Dynamic Dense | 10.83 | −R dense | 11.14 |
| +Mul. Dyn. Dense+Re-alloc | **10.77** | | |
| +Re-alloc | 11.93 | | |

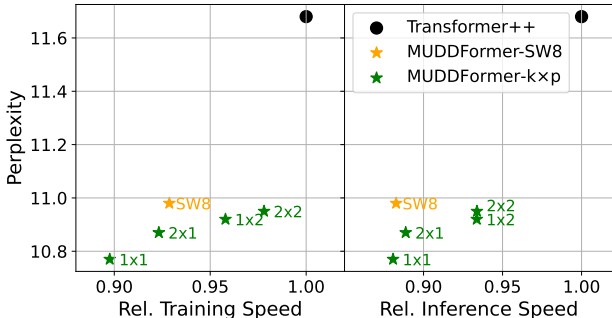

*Figure 9.* PPL vs. relative training and inference speed of MUD-DFormer variants.

**Variants with Sparse Connectivity** We design MUD-DFormer variants by approximating its dense connections with two sparse connectivity patterns: 1) *dilation and periodicity* (MUDDFormer-$k \times p$, also used in (Pagliardini et al., 2024)): each DA module aggregates the outputs of every $k$-th block, and the DA modules are inserted after every $p$ blocks. 2) *sliding window* (MUDDFormer-SW$n$): each DA module accesses to the outputs from only previous $n$ blocks plus the embeddings. Figure 9 shows the Pile validation perplexities and relative training and inference speed (compared to Transformer++) of these MUDDFormer variants. We measure perplexities with 405M models to align with ablation results in Table 5, while training and inference speeds are measured using 1.3B untrained models as in Table 4, the size of which is of more practical value.

While fully dense ($1 \times 1$) connectivity achieves the best performance, these sparse variants provide a spectrum of performance-efficiency trade-offs. For example, switching from MUDDFormer-$1 \times 1$ to MUDDFormer-$2 \times 2$ increases relative training/inference speed from 89.8%/88.1% to 97.8%/93.4% with only a 0.18 increase in ppl. In comparison, MUDDFormeer-SW8 is inferior with a lower speed / ppl ratio, highlighting the indispensability of long-range (>8) cross-layer interactions.

## 4. Related Work

**Enhancing Residual Connections** Despite the pervasive use of residual connections (He et al., 2016) in modern deep architectures, various approaches have been proposed to address their issues such as representational collapse and diminishing return for deeper models by strengthening cross-layer communication. Huang et al. (2017) introduced densely connected convolutional networks (DenseNet) for image classification. Inspired by DenseNet, Pagliardini et al. (2024) proposed DenseFormer for Decoder-only Transformers, which uses *Depth Weighted Averaging* modules to aggregate outputs from all preceding layers with static, learnable weights. Similarly, Wang et al. (2019) proposed Dynamic Linear Combination of Layers (DLCL) in deep

encoder-decoder Transformers for neural machine translation, which also used static and learnable dense connection weights. MUDD connections also have some linkage with HighwayNetworks (HN) (Srivastava et al., 2015). Both take inspiration from sequence model architectures and apply depthwise (HN from LSTM, MUDD from attention). HN is also the first to propose the critical concept of input dependent gating when mixing outputs between layers. Most recently, Zhu et al. (2025) proposed Hyper-Connections (HC), an alternative to residual connections that uses both static and dynamic weights to adjust inter-layer dependencies. Other research has explored different forms of cross-layer attention (ElNokrashy et al., 2022; Fang et al., 2023; Wang et al., 2024) which retrieve or update representations across different layers in a more flexible manner.

MUDDFormer is closely related to DenseFormer and HC but differs in critical ways. First, unlike DenseFormer, our MUDD connections *dynamically* compute per-position weights conditioned on the hidden states. Second, although HC uses a combination of static and dynamic weights to expand the hidden states, it does not employ explicit all-to-all cross-layer dense connectivity. Moreover, none of existing approaches consider decoupling the four input streams of a Transformer block by a *multiway* design, which is shown to bring significant performance gain in MUDDFormer.

**Mechanistic Interpretability** Research in this field employs various attribution methods (Conmy et al., 2023; Hanna et al., 2024) to uncover the circuits within Transformers that underlie specific capabilities (Elhage et al., 2020; Wang et al., 2024; Ni et al., 2025). These studies reveal the critical role of cross-layer interactions between attention heads and MLPs in enabling complex reasoning - a key insight motivating MUDD connections' design, which explicitly facilitates such interactions.

**Cross-Layer KV Cache Optimization** Brandon et al. (2024) proposes Cross-Layer Attention (CLA) to reduce Transformer KV cache size by sharing keys and values between adjacent layers, trading expressiveness for efficiency. Our MUDD connections enable cross-layer information flow between KV caches via dense connections on key and value streams. This enhances KV cache expressiveness and utility, improving in-context learning as evidenced by experiments. OmniNet (Tay et al., 2021a) achieves a fully global KV Cache receptive field by allowing each token to attend to *all* tokens in *all* layers. The authors reported that the additional omni (i.e. all-to-all) attention in OmniNet is computationally expensive and proposed to mitigate it by efficient attention variants. In contrast, MUDD connections is much more efficient because: 1) Only depth-wise aggregation (DA) is introduced, since the other omni attention paths can be formed by composition of DA and within-layer MHA; 2) DA is implemented as lightweight query-wise

attention. The computation overhead, both theoretical and practical, is much lower.

**Intra-Layer Architectural Innovations** Many other studies attempt to enhance the performance or efficiency of foundational sequence models with individual layers, including attention mechanisms (Liu et al., 2024a; Xiao et al., 2024a; Ye et al., 2024; Leviathan et al., 2024), sub-quadratic linear attention or RNN/SSM architectures (Gu & Dao, 2024; Dao & Gu, 2024; Peng et al., 2024; Yang et al., 2024) and sparse Mixture-of-Experts (MoE) (Fedus et al., 2022; Dai et al., 2024). By contrast, MUDD connections focus on *cross-layer* communication, making it orthogonal and complementary to these approaches. We leave the exploration of combining MUDD connections with these within-layer optimizations for future work.

## 5. Conclusion

We introduced Multiway Dynamic Dense connections to address the limitations of residual connections and enhance cross-layer information flow in Transformers. Experimental results showed that MUDDFormer is effective, efficient and scalable. It significantly outperforms Transformer baselines in language and vision tasks and improves emergent abilities such as in-context learning with minimal overhead. MUDD connections have the potential to become an indispensable component of next-generation foundation models.

## Acknowledgements

We are grateful to Google Cloud for providing the compute for model training, and to Shun Wang and Tingting Zhang for their technical support and help in troubleshooting TPU resource allocation and training.

## Impact Statement

Our work introduces Multiway Dynamic Dense (MUDD) connections, a lightweight architectural innovation that fundamentally enhances Transformer-based foundation models. MUDD connections achieve up to $2.4\times$ training efficiency gains while improving in-context learning capabilities, a critical requirement for real-world LLM applications. This breakthrough directly addresses the escalating computational and environmental costs of training ever-larger models, offering a sustainable pathway to high-performance AI without exponential parameter growth.

The open-source release of MUDDFormer architectures and pre-trained models will democratize access to state-of-the-art model efficiency techniques, particularly benefiting resource-constrained researchers and organizations. Enhanced attention mechanisms from MUDD's multiway design could advance mechanistic interpretability research

by producing more structured attention patterns and circuits.

While our method itself is architecture-focused, we acknowledge that more capable language models could potentially be misused for harmful content generation. However, MUDDFormer's efficiency gains may paradoxically mitigate this risk by reducing the energy barrier to developing safer, smaller models that match larger counterparts' performance. We commit to implementing rigorous model release protocols aligned with responsible AI practices.

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

## A. Dynamic Dense Connections as Depth-wise Self-Attention

Given a sequence $x \in \mathbb{R}^{T \times D}$, the output of dot-product self-attention at position $i$ is:

$$\text{softmax}(x_i W^Q (x_{:i} W^K)^T) x_{:i} W^V \tag{11}$$

where $x_{:i} W^K \in R^{(i+1) \times d}$ ($d$ is head dim) are $i + 1$ input-dependent keys. Combining Eq. (5) and Eq. (6) in Section 2, the output of dynamic DA at layer $i$ for position $t$ is (operating depthwise across layers):

$$(\text{GELU}(X_i[t] W_1) W_2 + a_i) X_{:i}[t] \tag{12}$$

Comparing Eq. (11) and Eq. (12), $W_1$ plays the role of query projection ($W^Q$) and $W_2^T \in R^{(i+1) \times d}$ ($d = i + 1$ is the inner dim of MLP, also the head dim) are $i + 1$ keys as parameters independent of input. So dynamic DA can be seen as lightweight self-attention except:

- Keys are independent of input;

- A learnable positional bias $a_i$ is used;

- Softmax is removed. Instead, GELU activation is applied to query (more like linear attention);

- $W^V$ transformation is not used.

While theoretically these simplifications may impact the representation capacity, we empirically found that adding more sophisticated ingredients in DA (e.g. input dependent keys, softmax) does not bring improvement and slow down training.

## B. PyTorch Style Pseudo-code for MUDDFormer

```
1   # B = batch_size;  T = seq_len; D = model_dim
2   # L = layer_index; C = num_ways = 4;  K = DA_hidden_dim = C*(L+1)
3
4   def generate_dw(x, mudd_theta): # x: BxTxD
5       w1, w2, a = mudd_theta   # w1: DxK, w2: Kx(C*(l+1)), a: Cx(l+1)
6       dw = GELU(RMSNorm(x) @ w1) @ w2 + a
7       dw = rearrange(dw, 'B T (C L)-> C B T L', C=4)
8       return dw
9
10  def DA(Xs, mudd_theta): # Xs: List (l+1)x[BxTxD]
11      dw = generate_dw(Xs[-1], mudd_theta)
12      xs = []
13      for c, way in enumerate(['Q', 'K', 'V', 'R']):
14          x = sum([dw[c, :, :, j:j+1] * Xs[j]   # BT1,BTD->BTD
15                  for j in range(len(Xs))])
16          xs.append(x)
17      return xs
18
19  def muddformer(x, model):
20      x = model.embedding(x)
21      Xs = [x]
22      xq, xk, xv, xr = x, x, x, x
23      for block in model.blocks:
24          attn_out = block.attn(LN(xq), LN(xk), LN(xv)) + xr
25          x = block.ffn(LN(attn_out)) + attn_out
26          Xs.append(x)
27          xq, xk, xv, xr = DA(Xs, block.mudd_theta)
28      return xr
```

## C. Details of Complexity Analysis

Compared to Transformer++, extra compute and parameters in MUDDFormer are introduced by DA modules, increasing from bottom layer to top layer, due to varied hidden dim $K_i$ of DA at layer i. To estimate the overhead, we calculate $R_{\Delta params}$ and $R_{\Delta FLOPs}$, the ratios of extra parameters and computation by analyzing an average middle layer in MUDDFormer. For this layer, the average hidden dimension of DA is $\overline{K} = 4(\overline{L} + 1) = 4(\frac{L+1}{2} + 1) = 2(L + 3)$. In a typical Transformer architecture we assume $D \gg \overline{K}$ and define $\rho = \frac{T}{D}, \eta = \frac{L+3}{D}$. We omit RMSNorm because it is negligible in terms of parameters and computation.

### Ratio of extra parameters

In a Transformer++ of $L$ layers and $D$ model dimension, the number of parameters is $12LD^2$ ($4LD^2$ for Attention and $8LD^2$ for FFN). In MUDDFormer, the layer $i$ adds $DK_i$ parameters in $W_1$ and $K_i^2$ in $W_2$ of DA, so the ratio of extra parameters is as follows.

$$R_{\Delta params} = \frac{\sum_{i=1}^{L}(\overbrace{DK_i}^{W_1} + \overbrace{K_i^2}^{W_2})}{12LD^2} \tag{13}$$

Approximating $K_i$ with $\overline{K}$ and assuming $D \gg \overline{K}$:

$$\begin{aligned}
R_{\Delta params} &\approx \frac{D\overline{K} + \overline{K}^2}{12D^2} \\
&\approx \frac{D\overline{K}}{12D^2} (\text{assume } D \gg \overline{K} \text{ and ignore } \overline{K}^2) \\
&= \frac{\overline{K}}{12D} \\
&= \frac{2(L+3)}{12D} (\text{recall } \overline{K} = 2(L+3)) \\
&= \frac{L+3}{6D} \\
&= \frac{\eta}{6} (\text{denote } \eta = \frac{L+3}{D})
\end{aligned} \tag{14}$$

Thus, the ratio of extra parameters scales linearly with the rectified depth/width ratio $\eta = \frac{L+3}{D}$.

### Ratio of extra FLOPs.

In a Transformer++ model, the pretraining FLOPs is $2LDT(12D + T)$ for a sequence of length $T$. At layer $i$ of MUDDFormer, the extra FLOPs includes $2TDK_i + 2TK_i^2$ for generating dynamic dense weights $A_{ij}$ (Eq. (6)) and $2TDK_i$ for depthwise aggregate (Eq. (5)).

$$R_{\Delta FLOPs} = \frac{\sum_{i=1}^{L}(\overbrace{2TDK_i + 2TK_i^2}^{\substack{\text{generate dense weight } A_{ij} \\ \text{Code Line 6}}} + \overbrace{2TDK_i}^{\substack{\text{Depthwise Aggregate} \\ \text{Code Line 13-15}}})}{2LDT(12D + T)} \tag{15}$$

Similarly, we consider an average layer of MUDDFormer for simplification, then the extra FLOPS becomes $4TD\overline{K} + 2T\overline{K}^2$.

$$
\begin{aligned}
R_{\Delta FLOPs} &\approx \frac{4TD\overline{K} + 2T\overline{K}^2}{2DT(12D + T)} \\
&= \frac{2D\overline{K} + \overline{K}^2}{D(12D + T)} (\text{divide } 2T) \\
&\approx \frac{2D\overline{K}}{D(12D + T)} \ (\text{assume } 2D \gg \overline{K} \text{ and ignore } \overline{K}^2) \\
&= \frac{4D(L + 3)}{12D^2 + DT} (\text{recall } \overline{K} = 2(L + 3)) \\
&= \frac{(L + 3)/D}{3 + T/4D} (\text{divide } 4D^2) \\
&= \frac{\eta}{3 + \rho/4} (\text{denote } \rho = \frac{T}{D}, \eta = \frac{L + 3}{D})
\end{aligned}
\tag{16}
$$

Therefore, the ratio is approximately $\frac{\eta}{3+\rho/4}$, decreasing with the rectified depth/width ratio.

## D. Hyperparameters and Baselines for Scaling Law Experiments

### D.1. Hyperparameters

We use the AdamW optimizer with $\beta_1 = 0.9$, $\beta_2 = 0.95$, gradient clip value of 1.0, weight decay of 0.1, 1% learning rate warmup steps followed by cosine decay to 10% of its maximal value, and no dropout. These hyperparameters are mostly taken from the GPT-3 paper (Brown et al., 2020) and are also used by all the baseline models listed below.

### D.2. Baseline Models

- **Transformer**: The standard Transformer based on GPT-3.

- **Transformer++**: An improved Transformer architecture adopted by Llama (Touvron et al., 2023) etc. with rotary positional encoding (RoPE) (Su et al., 2024), SwiGLU MLP (Shazeer, 2020), RMSNorm instead of LayerNorm and no linear bias.

- **DenseFormer**: The DenseFormer model from Pagliardini et al. (2024) without dilation, which has the best performance according to the paper. We implemented the model in JAX based on the PyTorch code released by the authors[8].

- **Hyper-Connections**: The dynamic hyper-connections with expansion rate $n = 4$ (DHC×4) from Zhu et al. (2025), which achieves superior results on language model pre-training and is the recommended configuration in the paper. We implemented the model in JAX based on the PyTorch Implementation given in Appendix J of the paper.

- **DDFormer**: Transformer with dynamic dense connections but without multiway splitting as described in Section 2.2.

## E. Memory Usage and Wall-Clock Time for Main Experiments

**Theoretical analysis of memory usage** In theory, peak activation memory usage for training a transformer in float16 with $L$ layers, $N$ heads, sequence length $T$, batch size $B$ and model dim $D$ occurs at the beginning of backpropagation and is composed of two parts:

1. hidden states for $L$ layers: $2LBTD$ (gradient checkpointing)

2. activation memory for a layer: $BTD(34 + 6NT/D)$ (outlined in (Korthikanti et al., 2023), recomputation of layer $L$ when backpropagate it)

---

[8]https://github.com/epfml/DenseFormer

We compare theoretical activation memory usages in Table 6. DenseFormer adds $2LBTD$ to store gradients for each layer's hidden state when backpropagating DA after layer $L$. Based on this, MUDDFormer adds another $6BTD$ for recomputation of layer $L$'s multi-way hidden states $Q,K,V$ when backpropagate it (they are not stored for all layers but are recomputed during backpropagation). The extra memory ratio for MUDD is $(L+3)/(L+17+3NT/D)$, and typical values are less than 30%. During inference, the activation memory usage is dominated by the $KV$ cache, which is not impacted by the extra memory brought by MUDD.

*Table 6.* Comparison of theoretical activation memory usages.

| model | activation memory | memory ratio |
|---|---|---|
| TFM++ | $2LBTD + BTD(34 + 6NT/D)$ | $1$ |
| DenseFormer | $2LBTD + BTD(34 + 6NT/D) + \mathbf{2LBTD}$ | $L/(L+17+3NT/D)$ |
| MUDDFM | $2LBTD + BTD(34 + 6NT/D) + \mathbf{6BTD} + \mathbf{2LBTD}$ | $(L+3)/(L+17+3NT/D)$ |

**Measured memory usage and wall-clock time** In Table 7, we report actual memory usage (measured using jax.profile) and wall-clock time for both the main (Figure 3, Table 3) and efficiency (Table 4) experiments. Besides activation memory, the actual memory usage also includes model parameters, gradients and optimizer states, and is affected by JAX compiler optimizations. For all model architectures and sizes, the relative training speed is 80%-90%, which could be further improved by custom kernel implementation. The extra memory ratio of MUDD is 20%-30%, comparable to that of HyperConnections (see Table 9 in their paper). As noted in our paper, the results for efficiency experiments (row 3-5) are of more practical relevance because they represent a more commonly used architecture (Transformer++) and model sizes.

*Table 7.* Comparison of measured activation memory usages and wall-clock time.

| model | model size | wall-clock time (hour) | rel. speed | mem (GB) | mem ratio | v5p pod size | tokens | batch size |
|---|---|---|---|---|---|---|---|---|
| TFM++ / MUDDFM | 405M | 5.7/7 | 81% | 86/111 | 29% | 16 | 7B | 0.5M |
| | 834M | 20/25.1 | 79% | 225/273 | 21% | 16 | 15B | 0.5M |
| | 1.4B | 29.5/32.5 | 91% | 301/386 | 28% | 32 | 26B | 0.5M |
| | 2.8B | 122/145 | 84% | 1352/1648 | 22% | 128 | 300B | 2M |
| | 6.9B | 251/262 | 96% | 1887/2222 | 17% | 128 | 300B | 2M |
| Pythia / MUDDPythia | 1.4B | 163/183 | 89% | 1296/1655 | 28% | 64 | 300B | 2M |
| | 2.8B | 124/154 | 81% | 1318/1655 | 25% | 128 | 300B | 2M |

# F. Image Classification with ViT

Besides decoder-only transformer for language modeling, we apply MUDD connections to Vision Transformer (ViT, an encoder-only Transformer) (Dosovitskiy et al., 2020) for image classification on the ImageNet-1k dataset (ILSVRC-2012). Implementation and experimental settings (e.g. the use of RandAugmention+MixUp, fixed 2D sincos position embedding and global average pooling) are based on the Big Vision codebase[9]. We use AdamW with $\beta_1 = 0.9$, $\beta_2 = 0.999$, gradient clip value of 1.0, weight decay of 0.3, learning rate of 0.003 with 10000 warmup steps followed by cosine decay to 0, batch size of 4096, RandAugment of 1evel 10, Mixup of 0.2 and dropout rate of 0.1. We use ViT-S/16 as the baseline model and equip it with MUDD connections to obtain MUDDViT-S/16. We also compare with a $1.72\times$ larger model ViT-M/16 (Table 8). We report validation loss and top-1 accuracy results on 90 and 300 epochs in Table 9. As can be seen, the gain from MUDD connections decreases a bit during the training progress, probably because many epochs of repeated passes over the same dataset diminish the additional expressive capacity brought by MUDD connections. Despite this, MUDDViT-S/16 still outperforms ViT-S/16 by 2% on epoch 300, also surpassing ViT-M/16.

*Table 8.* ViT Model architectures for ImageNet-1k classification.

| Model | $n_{layers}$ | $d_{model}$ | $d_{mlp}$ | $n_{heads}$ | params |
|---|---|---|---|---|---|
| (MUDD)ViT-S/16 | 12 | 384 | 1536 | 6 | 22M |
| ViT-M/16 | 12 | 512 | 2048 | 8 | 39M |

---

[9]https://github.com/google-research/big_vision

*Table 9.* ViT for ImageNet-1k classification results.

| Model | val. loss | acc@e90 | acc@e300 | Rel. size |
|---|---|---|---|---|
| ViT-S/16 | 0.993 | 53.4 | 76.0 | 1 |
| MUDDViT-S/16 | **0.871** | **56.9** | **78.1** | 1.007 |
| ViT-M/16 | 0.890 | 55.2 | 77.9 | 1.72 |

## G. Visualization

**Head activation from attention patterns** In Section 3.4, we show that the ratio of head activations in MUDDPythia-2.8B is larger than that in Pythia-2.8B. Here we draw the actual attention patterns on a randomly sampled sequence of length 32 from Pile validation set for the 32 heads in layer 25 of these two models in Figure 10 and 11. It is clear that attentions in Pythia mainly concentrate on the sink token (inactive) while attentions in MUDDPythia disperse on various tokens (active).

**Cross-layer dynamic weights** To better understand MUDDPythia, we visualize dynamic dense connection weights. Due to the high variance of the norm of hidden states $X_i$, we scale those weights by the average norm of each layer, rectifying the importance of weights. The rectified mean and standard deviation of dynamic connection weights in MUDDPythia-2.8B are shown in Figure 12 and 13, respectively. It is evident that the patterns of the four streams (query, key, value, residual) differ from each other, validating the necessity of separating them from the standard residual stream. It is noteworthy that in the value-stream connections, most layers have a salient and more dynamic weight on output of the first layer, thus forming a long-range channel to transport bottom information for attention heads in upper layers.

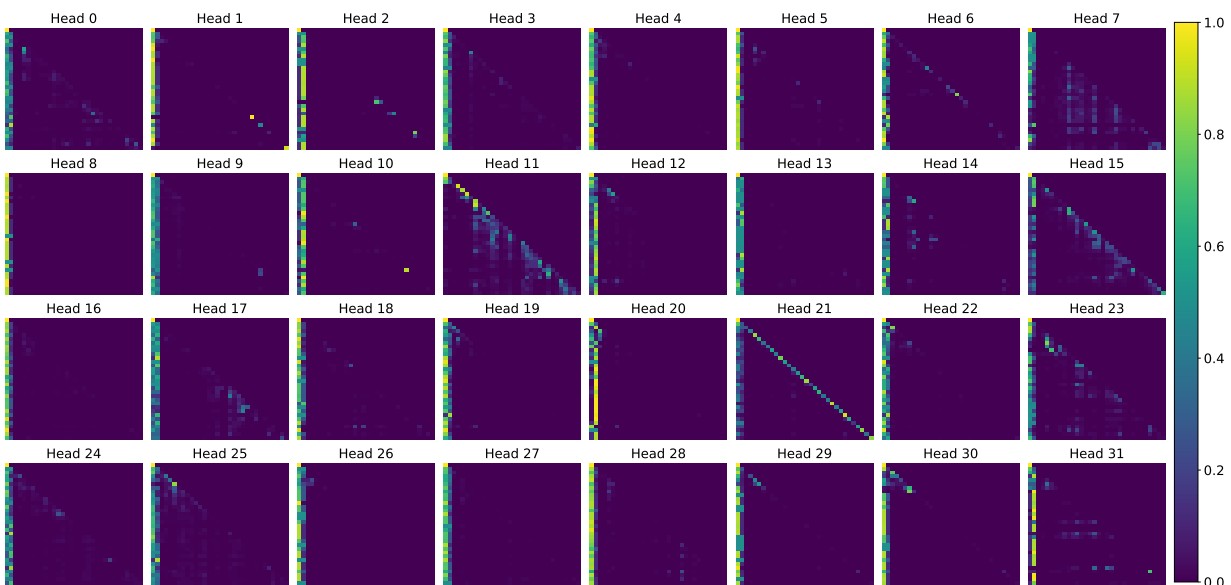

*Figure 10.* Attention patterns for the 32 heads in the 25th layer of Pythia-2.8B.

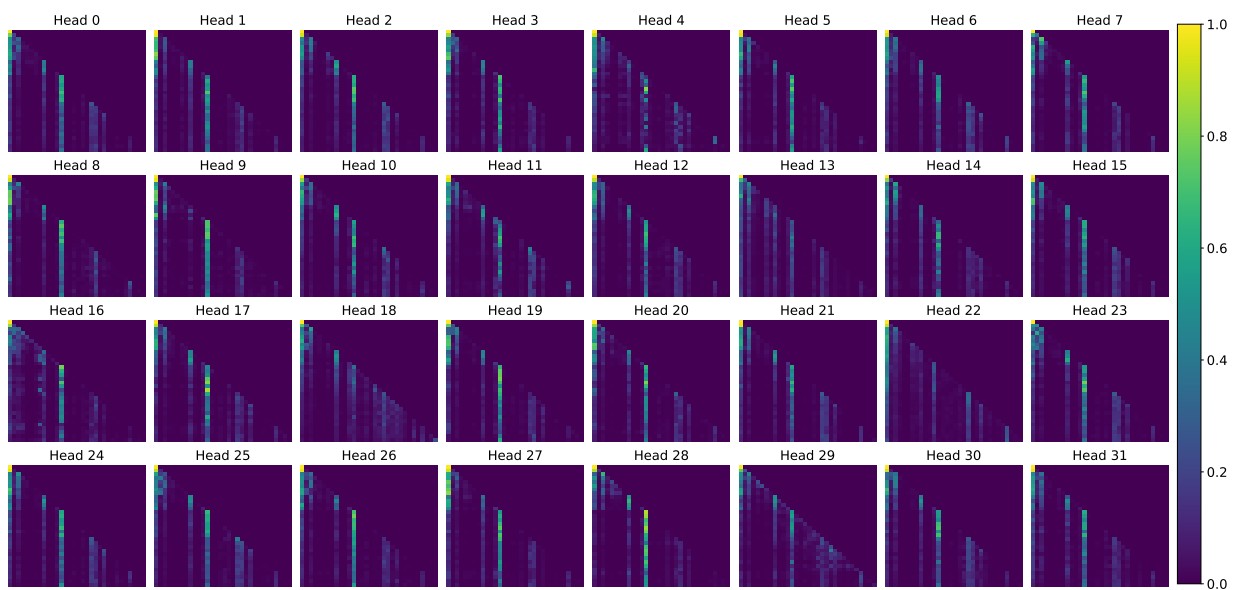

*Figure 11.* Attention patterns for the 32 heads in the 25th layer of MUDDPythia-2.8B.

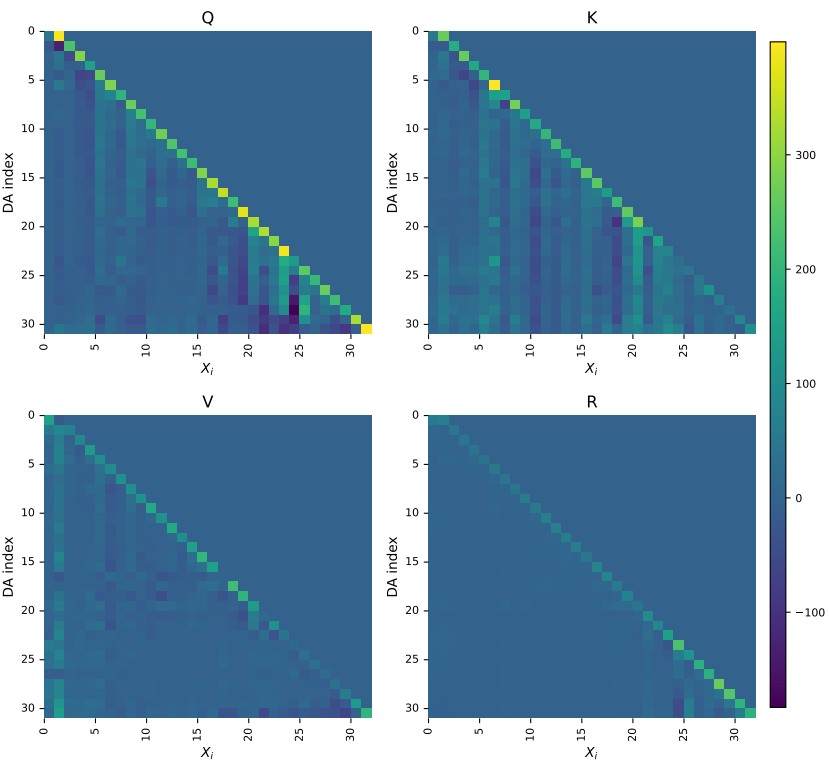

*Figure 12.* Mean of dynamic dense connections of MUDDPythia-2.8B.

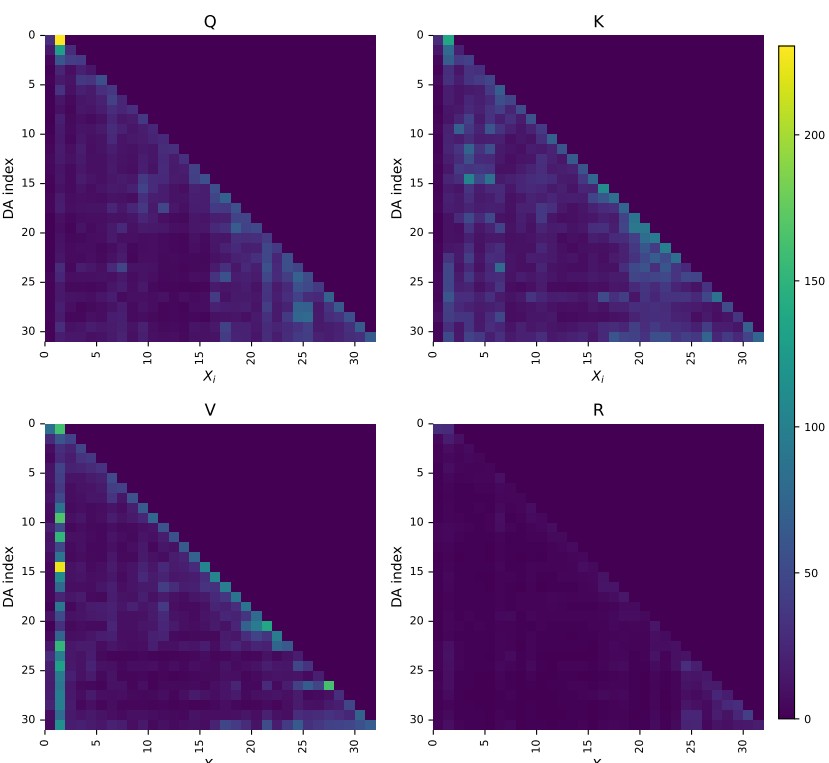

*Figure 13.* Standard deviation of dynamic dense connections of MUDDPythia-2.8B.

