# OpenReview forum: "MUDDFormer: Breaking Residual Bottlenecks in Transformers via Multiway Dynamic Dense Connections"
_ICML.cc/2025/Conference — ICML 2025 poster_

### Official Review · Reviewer_5mRp · 2025-03-10

**Overall Recommendation:** 3

**Summary:**

The paper proposes Multiway Dynamic Dense (MUDD) connections to improve cross-layer information flow in Transformers. By dynamically computing per-position weights for query, key, value, and residual streams, MUDD mitigates representation collapse and enhances depth efficiency.

**Claims And Evidence:**

Most claims are well-supported, but efficiency trade-offs and broader comparisons need discussion.

**Essential References Not Discussed:**

N/A

**Ethical Review Flag:**

Flag this paper for an ethics review.

**Experimental Designs Or Analyses:**

The experimental design is rigorous and supports the claims, though broader architectural comparisons and efficiency optimizations would strengthen the analysis.

**Methods And Evaluation Criteria:**

Methods and evaluation are well-aligned with the problem, but efficiency trade-offs and broader comparisons could strengthen the study.

Experiments confirm efficiency gains, depth utilization, and representation diversity.  However,  Inference slowdown (~10–15%) is measured (Table 4) but not addressed.  Focuses on DenseFormer and Hyper-Connections, lacking comparisons to OmniNet or MoE models.

**Other Comments Or Suggestions:**

N/A

**Other Strengths And Weaknesses:**

Strengths:
- The paper presents a novel combination of dense connections and dynamic weight generation, improving Transformer efficiency with minimal overhead.
- Strong empirical validation with extensive scaling laws, ablation studies, and interpretability analysis.
- Clear and well-organized writing with useful visualizations and pseudo-code for reproducibility.

Weaknesses:
- Inference slowdown (~10–15%) is measured but not optimized.
- Limited comparison to alternative architectures beyond DenseFormer and Hyper-Connections.
- No real-world deployment discussion, which would strengthen practical relevance.

**Questions For Authors:**

N/A

**Relation To Broader Scientific Literature:**

The paper builds on residual connections (He et al., 2016) and DenseFormer (Pagliardini et al., 2024), addressing Transformer depth inefficiencies (Merrill et al., 2022) and representation collapse (Liu et al., 2020b) with dynamic multiway dense connections, though it lacks comparisons to alternative cross-layer architectures like OmniNet (Tay et al., 2021a) or MoE-based models (Fedus et al., 2022).

**Theoretical Claims:**

The paper does not present formal theoretical claims or proofs. Its contributions focus on empirical improvements through MUDD connections rather than theoretical guarantees. The complexity analysis (Section 2.6) is straightforward and aligns with standard FLOP and parameter counting methods. No issues were found.

---

> ### Author Rebuttal · Authors · 2025-03-29
>
> Thanks!
>
> > it lacks comparisons to alternative cross-layer architectures like OmniNet (Tay et al., 2021a) or MoE-based models (Fedus et al., 2022).
>
> **vs. OmniNet**
>
> We've already compared MUDD connections with OmniNet briefly in Related Work. While both aim at promoting information flow across the whole model, they have major differences which we elaborate below.
>
> The additional Attend operation in OmniNet for omni (i.e. all-to-all) attention is computationally expensive (overhead generally larger than the whole original forward pass because of much longer seq length). The authors propose to mitigate it by efficient attention variants. MUDD connections is much more efficient by: 1) Only depth-wise aggregation (DA) is introduced, because the other omni attn paths can be formed by composition of DA and within-layer MHA; 2) DA is implemented as lightweight query-wise attention (pls see reply to RYaY). The computation overhead (both theoretical and practical) is much lower.
>
> There is no official open source implementation of OmniNet and we do not implement and evaluate it due to time constraints. Instead here we do a rough comparison based on the C4 language modeling results in Table 1 of their paper, because it is most similar to our main experiments in terms of task, data and training compute. They compare models with roughly the same number of parameters, as also done in our experiments.
> MUDDPythia-1.4B/2.8B's relative reduction on ppl (5.1%/5.1%, see Table 2) is larger than OmniNet_T (4.2%) with the most expensive Transformer self-attention, which further degrades to OmniNet_P's 2.6% and OmniNet_B's 0.4% when more practical efficient attention is used to mitigate the prohibitive computation cost of OmniNet_T.
>
> The distinctive advantage of OmniNet over MUDD is allowing early layers to attend to late layers (i.e. back attention). This recurrent processing can increase the effective depth. How to enable back attention while maintaining efficiency is a very valuable research direction which we'd like to explore in the future.
>
> **vs. MoE**
>
> MoE and MUDDFormer are both architectures with dynamic weights. However, MoE uses these weights to select experts *within* a layer while MUDDFormer uses the weights to aggregate outputs *across* layers. They are complementary approaches and can be combined. To empirically compare MoE and MUDDFormer, we train two MoE models MoE-1.8B-A405M and MoE-4B-A834M with the same activated parameters as 405M and 834M dense models, using the same training settings in the scaling law experiments. For MoE specific settings, we largely follow OLMoE, choose 2 experts out of 16, use token choice policy with expert capacity 1.5 and load balance loss coefficient 0.01. Although the FLOPs of MoE is larger than that of MUDDFormer because of expert capacity greater than 1, and the number of total parameters is 4.4x - 4.8x larger, MUDDFormer is slightly better than MoE in perplexity (table below), showing its efficient utilization of *both* parameter and computation. In contrast, MoE works by *decoupling* parameter and computation and relies on significantly expanding parameters for good performance.
>
> |total params|activated params|TFM++|TFM++ MoE|MUDDFormer
> |-|-|-|-|-|
> |405M / 1.8B / 405M |405M|11.69 |10.83|**10.77**
> |834M / 4B / 834M |834M|9.47 |8.88|**8.85**
>
> >Inference slowdown (~10–15%) is measured but not optimized.
>
> As discussed in the paper, inference slowdown primarily stems from the series of small operations. Currently we implement inference in pure PyTorch and optimize speed mainly relying on torch.compile. We believe it can be further optimized by writing custom kernels which is left for future work. We'd like to note that even without further optimization, the 10-15% slowdown is entirely worth paying in practice considering the 1.8x-2.4x gain in computation and parameter utilization.
>
> > Limited comparison to alternative architectures beyond DenseFormer and Hyper-Connections.
>
> As most recent cross-layer architectures, the authors of DenseFormer and Hyper-Connections (HC) already compare them with several related approaches and show their advantage over these approaches, e.g. DenseFormer > Depthwise Attention (ElNokrashy et al., 2022), HC > ResiDual (Xie et al., 2023), Altup (Baykal et al., 2024). Therefore, we do not compare with these cross-layer approaches anymore by showing MUDD connections > HC > Denseformer.
>
> > No real-world deployment discussion, which would strengthen practical relevance.
>
> As a general component that significantly enhance Transformer architecture, MUDD connections have broad real-world applications, especially for compute or memory resource-constrained settings, e.g. running LLMs on mobile, considering MUDDFormer's much improved parameter and computation utilization over Transformer. For example, one can use a ~2B MUDDFormer model to replace a twice bigger Transformer model and still get similar performance, so that the resource requirement can be lowered,

---

### Official Review · Reviewer_RYaY · 2025-03-13

**Overall Recommendation:** 3

**Summary:**

The paper proposes a new method for aggregating information from previous layers that improves information flow in transformers. Specifically, the authors combine two methods for the aggregation: gated, token-specific weighting and separate streams for inputs to the attention (queries, keys, values, and residuals). The authors demonstrate improvements on the transformer models with fixed FLOPs per training, when compared to models not incorporating the MUDD framework.

**Claims And Evidence:**

The authors claim that adding multiway dynamic dense (MUDD) connections improves the performance of the model. They demonstrate this on the example of the transformer architecture by integrating MUDD and showing that it can achieve the performance of models trained with twice the theoretical compute.

While the evidence in terms of theoretical compute is very strong, the paper would benefit from reporting additional empirical measurements of performance from the main training, such as memory usage and wall-clock time for the main experiments. Aggregating multiple layers might lead to increased memory consumption, so it is important to consider this while designing a scalable architecture. Moreover, there are discrepancies between theoretical FLOP computations and practical wall-clock time. While the paper reports tokens/s in Table 4, this metric reflects latency for different models than those compared in the main claim with scaling curves. The paper would be improved if the timing was shown for the main experiment or if both were provided.

**Essential References Not Discussed:**

I did not find any essential references not discussed in the paper.

**Experimental Designs Or Analyses:**

The experimental comparison in the fixed FLOP scenario, particularly in relation to Pythia, makes sense. There are additional analyses that further give intuition about the underlying mechanisms in MUDD.

**Methods And Evaluation Criteria:**

The selected benchmarking datasets and evaluation criteria are sensible but should be extended to include memory and wall-clock time measurements for the main experiments.

**Other Comments Or Suggestions:**

-

**Other Strengths And Weaknesses:**

The paper would benefit from additional clarity and motivation in some sections, and figures could be described more thoroughly. Some claims might be refined or supported with further justification. For instance:

- l14-l17: The discussion of how Pre-Norm stabilizes training by preventing representation collapse (Liu et al., 2020b) - the citation does not support this claim and the claim itself is not related to the paper.
- l105-108: presenting dynamic dense connections as depth-wise single-headed self-attention needs further justification.

Additionally, the paper is full of grammatical errors and wording issues:

- l078: "benefit from differentiated" -> "benefit from different.", also 'different' as a word here is redundant because of what follows directly after.
- l088: "differentially" -> "differently."
- l138: "grows" -> "grow.", l640: "calculates" -> "calculate" (this error appears in multiple locations).
- l643: "negible"
etc.

**Questions For Authors:**

In summary, while the paper contains several minor issues that, if addressed, could further improve its clarity and impact, the main result is both strong and potentially impactful, particularly from a scalability perspective. I am willing to reconsider my evaluation once these issues have been addressed.

**Relation To Broader Scientific Literature:**

The paper effectively positions itself as a continuation of research on DenseNet and DenseFormer by introducing dynamic connections to improve selectiveness in aggregation. It covers relevant recent literature, and it may be beneficial to also cite HighwayNetworks [1], which introduced a gated version of the residual connection earlier. This citation could further enrich the paper’s motivation by linking it to foundational ideas that combine aspects of DenseNet with the original HighwayNetworks concepts.

[1] Srivastava, Rupesh Kumar, Klaus Greff, and Jürgen Schmidhuber. "Highway networks." arXiv preprint arXiv:1505.00387 (2015).

**Theoretical Claims:**

The only theoretical claim is the complexity analysis presented in Appendix B. However, the transitions in the analysis (equations 11 and 12) are extremely informal and could be clarified further with a more detailed derivation.

---

> ### Author Rebuttal · Authors · 2025-03-29
>
> Thanks! (mem = memory, backprop = backpropagating)
>
> >mem usage and wall-clock time for main experiments
>
> Peak activation mem usage for training a transformer in float16 with L layers, N heads, seq length T, batch size B and model dim D occurs at the beginning of backprop and is composed of two parts:
> - hidden states for L layers: 2LBTD  (grad checkpointing)
> - activation mem for a layer: BTD(34+6NT/D)  (outlined in [1], recomputation of layer L when backprop it)
>
> Comparison of activation mem usages:
> |model|activation mem|mem $\Delta$ ratio|
> |-|-|-|
> |TFM++|2LBTD+BTD(34+6NT/D)|1|
> |DenseFormer|2LBTD+BTD(34+6NT/D)+**2LBTD**|L/(L+17+3NT/D)|
> |MUDDFM|2LBTD+BTD(34+6NT/D)+**6BTD**+**2LBTD**|(L+3)/(L+17+3NT/D)|
>
> DenseFormer adds 2LBTD to store gradients for each layer's hidden state when backprop DA after layer L. Based on this, MUDDFormer adds another 6BTD for recomputation of layer L's multi-way hidden states Q,K,V when backprop it (they are not stored for all layers but are recomputed during backprop). The extra memory ratio for MUDD is (L+3)/(L+17+3NT/D) and typical values are less than 30%. During inference, the activation mem usage is dominated by the KV cache, which is not impacted by the extra mem brought by MUDD.
>
> We report actual mem usage (measured using jax.profile) and wall-clock time for both the main (Fig 2, Tab 3) and efficiency (Tab 4) experiments below. Beside activation mem, the actual mem usage also includes model params, grads and optimizer states, and is affected by JAX compiler optimizations.
> |model|model size|wall-clock time (hour)| rel. speed |mem (GB)|mem $\Delta$ ratio|v5p pod size|tokens|batch size|
> |-|-|-|-|-|-|-|-|-|
> |TFM++ / MUDDFM|405M|5.7/7|81%|86/111|29%|16|7B|.5M|
> ||834M|20/25.1|79%|225/273|21%|16|15B|.5M|
> ||1.4B|29.5/32.5|91%|301/386|28%|32|26B|.5M|
> ||2.8B|122/145|84%|1352/1648|22%|128|300B|2M|
> ||6.9B|251/262|96%|1887/2222|17%|128|300B|2M|
> |Pythia / MUDDPythia|1.4B|163/183|89%|1296/1655|28%|64|300B|2M|
> ||2.8B|124/154|81%|1318/1655|25%|128|300B|2M|
>
> For all model architectures and sizes, the relative training speed is ~80%-90%, which could be further improved by custom kernel implementation. The extra memory ratio of MUDD is ~20%-30%, comparable to that of HyperConnections (Tab 9 in their paper). As noted in our paper, the results for efficiency experiments (row 3-5) are of more practical relevance because they represent more commonly used architecture (Transformer++) and model sizes.
>
> [1] Korthikanti et al. (2022). Reducing activation recomputation in large transformer models
>
> > the analysis (eq. 11 and 12) could be clarified further with a more detailed derivation.
>
> Due to space limit, we'll refine the derivation and add explanation in the revised version.
>
> >citation to HighwayNetworks
>
> As the precursor to residual connections, HighwayNetworks (HN) indeed has some fundamental linkage with MUDD connections, e.g. both take inspiration from sequence model architectures and apply depthwise (HN from LSTM, MUDD from attention). HN is also the first to propose the critical concept of *input dependent* gating when mixing outputs between layers. We'll cite HN in the revised version and discuss the linkage with MUDD in more detail.
>
> >The discussion of how Pre-Norm stabilizes training by preventing representation collapse (Liu et al., 2020b)
>
> There may be a misunderstanding. What Liu et al. propose is that pre-ln *causes*, rather than prevents, representation collapse (RC) (see the paragraph before Section 4.2 in their paper). Some other papers (e.g. Hyper-Connections, ResiDual) also cite Liu et al. when discussing RC. Since RC in deep pre-ln Transformers is one of the issues tackled by MUDD, we cite this paper too.
>
> >dynamic dense connections as depth-wise self-attention needs further justification.
>
> Given a sequence $x\in R^{T\times D}$, the output of dot-product self-attention at position i is:
>
> $softmax(x_iW^Q(x_{:i}W^K)^T)x_{:i}W^V$ (a)
>
> where $x_{:i}W^K\in R^{(i+1)\times d}$ (d is head dim) are i+1 input-dependent keys.
>
> Combining (5) and (6) in the paper, the output of dynamic DA at layer i for position t is (operating depthwise across layers):
>
> $(GELU(X_i[t]W_1)W_2+a_i)X_{:i}[t]$ (b)
>
> Comparing (a) and (b), $W_1$ plays the role of query projection ($W^Q$) and $W_2^T\in R^{(i+1)\times d}$ (d=i+1 is the inner dim of MLP, also the head dim) are i+1 keys as parameters independent of input. So dynamic DA can be seen as lightweight self-attention except:
> - keys are independent of input;
> - a learnable positional bias $a_i$ is used;
> - softmax is removed. Instead, GELU activation is applied to query (more like linear attention);
> - $W^V$ transformation is not used.
>
> While theoretically these simplifications may impact the representation capacity, we empirically found that adding more sophisticated ingredients in DA (e.g. input dependent keys, softmax) does not bring improvement and slow down training.
>
> >grammatical errors and wording issues
>
> Thank you! We'll fix them in the revised version.

---

> > ### Comment · Reviewer_RYaY · 2025-04-04
> >
> > I would like to thank the authors for the comments. Given the authors comments i decided to increase my score. The biggest concern left for me is that for now the method offers a trade-off between memory+wall-clock time and perplexity. It would be beneficial to demonstrate that MUDD can create configurations that at the same time limit memory, wall-clock time and have better perplexity than baselines (for example by limiting the number of parameters for MUDDformers). While current perplexity benefits are good, 30% of memory increase is also not negligible.

---

> > > ### Author Response · Authors · 2025-04-08
> > >
> > > >It would be beneficial to demonstrate that MUDD can create configurations that at the same time limit memory, wall-clock time and have better perplexity than baselines (for example by limiting the number of parameters for MUDDformers).
> > >
> > > We train a MUDDFormer using the same hyperparameters as training the 405M models in scaling law experiments, except reducing the number of layers from 24 to 19 to limit its number of parameters. As shown in the table below, compare with Transfomer++-405M, this model trains 4.4% faster, consumes 5.9% more memory, while still has lower perplexity.
> > >
> > > | model| n_layers | wall-clock time (hour) |rel. speed|mem (GB)|mem $\Delta$ ratio|ppl|
> > > |-|-|-|-|-|-|-|
> > > Transfomer++-405M | 24 | 5.7 | 100% | 86 | - | 11.68 |
> > > MUDDFormer-342M | 19 | 5.46 | 104.4% | 91 | 5.9% |11.12 |
> > >
> > > We'd like to emphasize two things:
> > > 1) **Interpreting this result**: Due to insufficient time, we only experiment with the ~400M sized models (81% relative training speed and 29% $\Delta$mem  for MUDDFormer) to show the *existence* of such a configuration. Based on the results in the table in our previous reply, for other larger model sizes, e.g. 6.9B (96% relative training speed and 17% $\Delta$mem), we may reduce fewer parameters for MUDDFormer, and thus sacrifice less performance, to approximate the wall-clock time and memory of the Transformer++ baseline.
> > > 2) **Wall-clock time vs memory usage**: While you put a strong emphasis on memory usage and we have empirically confirmed that MUDDFormer's actual memory usage is comparable to those of other cross-layer approaches as baselines (DenseFormer, Hyper-Connections), we still argue that training throughput (or equivalently wall-clock time per step) is generally more important than memory usage for model training. In a matched total wall-clock time setting, an architecture with higher training throughput (e.g. the baseline Transformer++) can train more steps and naturally get better results. On the other hand, most of our training runs on TPU are compute-bound (by total FLOPS of tensor cores) rather than memory bound (by HBM memory capacity), which means that with a given model size, modest increase of memory usage has little impact on training throughput as measured in tokens/second, though it can affect the max batch size that can be used. As known, larger batch size doesn't always lead to better results.

---

### Official Review · Reviewer_9FeX · 2025-03-17

**Overall Recommendation:** 3

**Summary:**

The paper proposes Multiway Dynamic Dense (MUDD) connections to enhance cross-layer information flow in Transformers by augmenting residual connections with dynamic, multi-headed dense aggregations. Key innovations include decoupling the query, key, value, and residual streams of each Transformer block, enabling dynamic weighting of connections based on position-dependent hidden states. This improves expressiveness over static/dense-only approaches (e.g., DenseFormer) and boosts scalability. MUDDFormer matches performance of larger models (e.g., Pythia-6.9B) with 2.4× less compute, achieving state-of-the-art results in language modeling while adding minimal overhead (~0.2% params, ~0.4% FLOPs). Ablations and analyses (e.g., attention activation, representation collapse) validate the method’s effectiveness.

**Claims And Evidence:**

The claims include:

MUDD improves Transformer scalability and efficiency by addressing residual stream bottlenecks.
Supported by experiments showing better perplexity curves vs. baselines (e.g., 834M MUDD matches 1.8× compute-to-loss of Transformer++).
Multiway dense connections enhance in-context learning (ICL).
Evidenced by 5-shot performance rivaling 12B models and analysis of attention head activation.
Minimal overhead (0.2% params, 0.4% FLOPs).
Verified in Section 2.6 and Table 1, though real-world overheads (Table 4) are slightly higher.
Potential gaps:

Claims about "circuit composability" rely on indirect analyses (e.g., head activation) rather than explicit mechanistic studies.
Efficiency claims could vary with hardware/implementation (real-world overheads are ~10-15% for training).

**Essential References Not Discussed:**

Limited discussion on other dynamic weight methods (e.g., Llama 3’s routing) or global memory approaches (OmniNet).

**Experimental Designs Or Analyses:**

Scaling laws span 405M to 1.4B models, with fair comparisons to Transformer++/Pythia.

Ablations isolate dynamic/multiway contributions.

Downstream evaluations cover both generative and ICL tasks, though tests on vision models are limited.

**Methods And Evaluation Criteria:**

The dense connection mechanism (dynamic, multiway) is well-motivated for addressing depth-related issues. Evaluation combines scaling laws, downstream tasks, and structural analyses. Benchmarks (Pile, FLAN) and baselines (Pythia, DenseFormer) are appropriate. However, results could benefit from more diverse tasks (e.g., structured vs. semi-structured data).

**Other Comments Or Suggestions:**

1. Consider releasing pretrained models for reproducibility.

**Other Strengths And Weaknesses:**

Strengths:

1. Simple yet effective method with minimal overhead.
2. Strong empirical results across scales.
3. Insightful analyses on representation collapse and head activation.
4. Extend experiments on imagenet.

Weaknesses:

1. Limited theoretical understanding of dynamic connections’ role.
2. Overheads could be prohibitive for edge/hardware-constrained scenarios.

**Questions For Authors:**

1. How does MUDD handle long contexts (e.g., 8k tokens)? Does it exacerbate memory issues?

**Relation To Broader Scientific Literature:**

Builds on DenseFormer/Hyper-Connections but innovates via dynamic weights and multiway streams.

**Theoretical Claims:**

No strict Theoretical Claims. This paper relies on empirical validation.

---

> ### Author Rebuttal · Authors · 2025-03-27
>
> Thanks!
>
> >Limited discussion on other dynamic weight methods (e.g., Llama 3’s routing) or global memory approaches (OmniNet).
>
> Although we don't know what you mean by "Llama 3's routing", we do a comparison with another representative dynamic weight method MoE, and a global memory approach OmniNet (which in fact has already been discussed briefly in Related Work). For detail please see reply to 5mRp.
>
> >Limited theoretical understanding of dynamic connections’ role.
>
> In order to get a better understanding of dynamic connections’ role, we extend the discussions in the paper on connections between MUDD and self-attention.
>
> **dynamic dense connections as single-head self-attention**
>
> Given a sequence $x\in R^{T\times D}$, the output of dot-product self-attention at position i is:
>
> $softmax(x_iW^Q(x_{:i}W^K)^T)x_{:i}W^V$ (a)
>
> where $x_{:i}W^K\in R^{(i+1)\times d}$ (d is head dim) are i+1 input-dependent keys.
>
> Combining (5) and (6) in the paper, the output of dynamic DA at layer i for position t is (operating depthwise across layers):
>
> $(GELU(X_i[t]W_1)W_2+a_i)X_{:i}[t]$ (b)
>
> Comparing (a) and (b), $W_1$ plays the role of query projection ($W^Q$) and $W_2^T\in R^{(i+1)\times d}$ (d=i+1 is the inner dim of MLP and is also the head dim) are i+1 keys as parameters independent of input. So dynamic DA can be seen as lightweight self-attention except:
> - keys are independent of input;
> - a learnable positional bias $a_i$ is used;
> - softmax is removed. Instead, GELU activation is applied to query (more like linear attention);
> - $W^V$ transformation is not used.
>
> While theoretically these simplifications may impact the representation capacity, we empirically found that adding more sophisticated ingredients in DA (e.g. input dependent keys, softmax) do not bring improvement and slow down training.
>
> **multiway dynamic dense connections as multi-head self-attention**
>
> For aggregating each of the four input streams (Q, K, V, R) there is a head. In our implementation (see Pseudo-code in Appendix A), the four heads share the query projection $W^Q$ while each head has its own set of keys $W_2^T$. The head dim is $d$ expanded four times.
>
> While there is no more theoretical analysis, there is an intuitive understanding of dynamic connections’ role (just as pointed out in the original Transformer paper (Vaswani et, al., 2017) in "Why Self-Attention"): as the role the horizontal attention over sequence in MHA plays, the depth-wise attention over layers in MUDD creates a *direct* path between any pair of layers, besides *mediated* through the shared residual stream which may get crowded when the model gets deeper.
>
> >Overheads could be prohibitive for edge/hardware-constrained scenarios.
>
> Considering its significant improvement over Transformer as shown in extensive experiments, one can in principle use a MUDDFormer model to replace a twice bigger Transformer model and still get similar performance, so the resource requirement is actually *lower*, making MUDDFormer suitable for edge/hardware-constrained scenarios. Nevertheless, we admit that deploying LLMs on edge devices is beyond our expertise and the overhead on them needs further verification. We'd like to know how you make this speculation. We could discuss further if you have specific concerns.
>
> >Consider releasing pretrained models for reproducibility.
>
> Sure. We'll release training and inference code in both JAX and PyTorch and the pretrained models once the paper gets accepted.
>
> >How does MUDD handle long contexts (e.g., 8k tokens)? Does it exacerbate memory issues?
>
> For training a model with L layers, N heads, seq length T, batch size B and model dim D, the extra activation memory ratio for MUDDFormer over Transformer is (L+3)/(L+17+3NT/D) (please see reply to RYaY for derivation), which decreases as context length T increases. We measure and compare the actual memory usage for training MUDDFormer-6.9B and Transformer++-6.9B on 8K tokens of contexts, using the same batch size of 2M tokens as the efficiency experiment in Table 4. The extra memory usage ratio of MUDDFormer is ~10%, which would not become the bottleneck or cause memory issues under normal circumstances. For inference, the activation mem usage is dominated by the KV cache which grows with context length, so the relative impact of MUDD connections on memory consumption diminishes as the context gets longer.

---

### Decision · Program_Chairs · 2025-05-01

**Decision:**

Accept (poster)

**Comment:**

I recommend accepting this paper. The proposed MUDD connections enhance Transformers by dynamically weighting cross-layer connections for each input stream. All three reviewers gave weak accept ratings. The method shows good results, matching Transformers trained with 1.8x-2.4x compute while adding only very few parameters. Despite some efficiency tradeoffs, the significant performance improvements and strong scaling properties across model sizes make this a valuable contribution with practical applications, particularly for resource-constrained settings.